# Water Cycle Algorithm for Modelling of Fermentation Processes

**Olympia Roeva** [1] **, Maria Angelova** [2] **, Dafina Zoteva** [1,3] **and Tania Pencheva** [2,*]

[1] Department of Bioinformatics and Mathematical Modelling, Institute of Biophysics and Biomedical Engineering, Bulgarian Academy of Sciences, 1113 Sofia, Bulgaria; olympia@biomed.bas.bg (O.R.); dafy.zoteva@gmail.com (D.Z.)

[2] Department of QSAR and Molecular Modelling, Institute of Biophysics and Biomedical Engineering, Bulgarian Academy of Sciences, 1113 Sofia, Bulgaria; maria.angelova@biomed.bas.bg

[3] Department of Computer Informatics, Faculty of Mathematics and Informatics, Sofia University "St. Kliment Ohridski", 1164 Sofia, Bulgaria

* Correspondence: tania.pencheva@biomed.bas.bg

**Abstract:** The water cycle algorithm (WCA), which is a metaheuristic method inspired by the movements of rivers and streams towards the sea in nature, has been adapted and applied here for the first time for solving such a challenging problem as the parameter identification of fermentation process (FP) models. Bacteria and yeast are chosen as representatives of FP models that are subjected to parameter identification due to their impact in different industrial fields. In addition, WCA is considered in comparison with the genetic algorithm (GA), which is another population-based technique that has been proved to be a promising alternative of conventional optimisation methods. The obtained results have been thoroughly analysed in order to outline the advantages and disadvantages of each algorithm when solving such a complicated real-world task. A discussion and a comparative analysis of both metaheuristic algorithms reveal the impact of WCA on model identification problems and show that the newly applied WCA outperforms GA with regard to the model accuracy.

**Keywords:** water cycle algorithm; genetic algorithm; parameter identification; fed-batch fermentation processes

## 1. Introduction

When studying complex systems such as biological ones, the interactions and behaviour of their components, i.e., molecules, cells, organs, and organisms, are explored in order to achieve their better understanding. Examples of biological systems on a macro scale are populations of organisms. Going deeper in the hierarchy, populations of microorganisms are in focus when biotechnological processes, and in particular, fermentation processes (FP), are considered. In the last decades, the investigation of FP has been a research area of unceasing interest due to their widespread applicability in the industrial production of medicines, vaccines, alternative biofuels, agro-based foods such as bread, beer, wine, etc. [1]. Among the most widely used microorganisms are bacteria and yeast due to their significant industrial and economic value. They both are also model organisms in genetics, used as synthetic pathways for antibiotics and other biomolecules of interest. As the most extensively studied model organisms, *Escherichia coli* is chosen here as a representative of the bacteria and *Saccharomyces cerevisiae*—of the yeast. The scientific efforts are directed to a deeper understanding of cellular growth control mechanisms in order to overcome several technological barriers and to achieve efficient and cost-effective bioprocesses [2].

FP have numerous specific features, since they combine biological and non-biological processes. FP exhibit differently in nature interactions and reactions: (1) purely physical (stirring, heating, bubbling); (2) chemical and physicochemical (decomposition of chemical compounds, formation of products); (3) purely biological (growth and death of microorganisms); (4) biochemical (transformation of substrates), which makes their modelling even harder than modelling biological systems. The complex structure of FP models is usually described by a set of nonlinear differential equations with time-varying, tightly related process variables [3]. As such, their modelling is not a trivial problem to be solved, and the choice of a suitable optimisation method is essential for the successful identification of model parameters [4,5].

The main advantage of metaheuristic techniques, which have been proven as a good alternative to conventional optimisation methods, is that they find a satisfactory solution for a reasonable computational time [6]. Two categories of metaheuristics are known: Single-based and population-based solution methods [7]. Some of the most powerful nature-inspired metaheuristics have been developed and tested for solving different optimisation problems. Genetic algorithms (GA) [8,9], ant colony optimisation [10], artificial bee colony optimisation [11], bat algorithm [12], and cuckoo search [13] have been proved successful in the parameter identification of FP models [14–18], structural optimisation problems [19], etc.

Since optimisation is the process of making something as good as possible, researchers continuously seek new metaheuristic methods that outperform the existing ones. In 2012, Eskandar et al. proposed a promising metaheuristic approach called water cycle algorithm (WCA) [20]. The inspiration for WCA was derived from the natural world, after observation of the entire hydrological cycle, including the flow of streams and rivers into the seas. Different physical processes are involved in this complex system, e.g., evaporation, condensation, precipitation, and surface run-off [20,21]. In nature, the sources of rivers are usually found in mountains as a result of snow melting. Rain water and other smaller tributaries sustain the rivers on their way from higher to lower places, before they subsequently flow into a sea, which is the most low-altitude place in the world. Underground water seeps into the permeable rock or soil layer and flows beneath the ground surface. Finally, water leaves the underground reservoir in springs or seeps. Meanwhile, evaporation occurs on the surface of oceans, lakes, rivers, and other water sources. During the photosynthesis process, plants transpire water. Evaporated water goes up into the cold air atmosphere and condenses in clouds. The water circulates and through the rain comes back to the earth again. Thus, the hydrological cycle closes.

WCA became popular due to the algorithm's ability to produce a better optimal solution while rapidly converging [21]. In a short time frame, WCA was successfully employed to various problems, such as the optimal operation of reservoir systems [22], attribute reduction problem [23], optimal cost design [24], detection of optimum reactive power dispatch problems [25], etc. The superiority of WCA in comparison to other considered methods is shown in [26,27] and is the reason why the algorithm keeps attracting more and more researchers' interest. Recently, WCA has been utilised to solve optimal power flow problems in electric grids [28], to enhance density grid-based clustering [29], to evaluate the optimum parameter values of a Proportional-integral-derivative (PID) controller for an automatic voltage regulator [30], etc.

The numerous aforementioned successful WCA applications have motivated the authors to adapt and apply WCA for the first time for solving the challenging problem of parameter identification of nonlinear FP models. Two different processes are chosen as subjects of this investigation: The bacteria *E. coli* and the yeast *S. cerevisiae* fed-batch FP. A comparison between the widely applied, nature-inspired, metaheuristic technique GA, presented for the first time by Holland [8], and WCA is further performed. GA is chosen as a referent metaheuristic technique since it has been proved as a successful optimisation method for solving different real-world problems [31], as well as for the parameter identification of yeast and bacteria FP models [14–18].

## 2. Materials and Methods

### *2.1. Fermentation Processes*

An experimental dataset of an *E. coli* fed-batch fermentation process has been conducted in the Institute of Technical Chemistry, University of Hannover, Germany [3]. The available datasets consist of on-line measurements of substrate (glucose) and off-line measurements of biomass.

#### 2.1.1. *Escherichia Coli* Fed-Batch Fermentation Process

In order to determine the glucose concentration on-line, a flow injection analysis (FIA) system was employed, using two pumps (ACCU FM40, SciLog, Saint. Louis, MO, USA) for constant sample and carrier injection at flow rates of 0.5 mL·min$^{-1}$ and 1.7 mL·min$^{-1}$, respectively. Cells containing culture broth (24 mL) were injected into the carrier stream and mixed with an enzyme solution of 350,000 U·L$^{-1}$ glucose oxidase (Fluka, Seelze, Germany) of a volume of 36 mL. CAFCA software (ANASYSCON, Hannover, Germany) was applied for the automation of the FIA system as well as for determining the glucose concentration. To reduce the measurement noise, the continuous-discrete extended Kalman filter was used.

Off-line samples were collected and rapidly centrifuged to analyse off-line the biomass dry weight (Shimadzu GC-14B, Duisburg, Germany) concentration. Samples of about 10 mL were taken roughly every hour. Off-line measurements were performed by using the Yellow Springs Analyser (Yellow Springs Instruments, Yellow Springs, OH, USA).

Figure 1 presents the feeding profile for the *E. coli* fed-batch FP considered here.

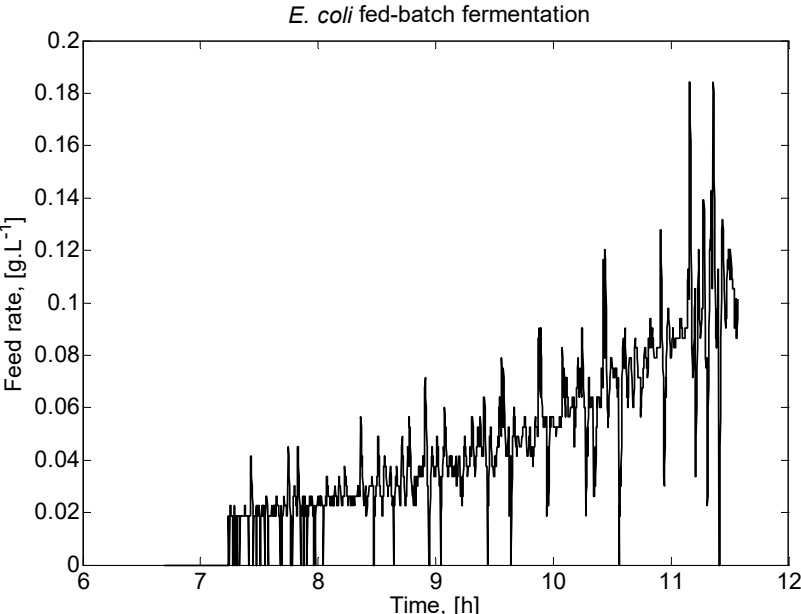

**Figure 1.** *E. coli* fed-batch fermentation process (FP) feeding profile.

A full description of the process conditions and experimental data of the fermentation are given in [3].

The *E. coli* fed-batch FP is described by a system of three differential equations modelling the nonlinear dynamics of the biomass and substrate [3]:

$$\frac{dX}{dt} = \mu_{max}\frac{S}{S+k_S}X - \frac{F}{V}X \tag{1}$$

$$\frac{dS}{dt} = -\frac{\mu_{max}}{Y_{S/X}}\frac{S}{S+k_S}X + \frac{F}{V}(S_{in} - S) \tag{2}$$

$$\frac{dV}{dt} = F \tag{3}$$

where $X$ denotes biomass concentration, $[g \cdot L^{-1}]$, $S$ denotes substrate concentration (glucose), $[g \cdot L^{-1}]$; $F$ denotes feeding rate, $[h^{-1}]$; $V$ denotes bioreactor volume, $[L]$; $S_{in}$ denotes initial glucose concentration in the feeding solution, $[g \cdot L^{-1}]$; $\mu_{max}$ denotes the maximum growth rate, $[h^{-1}]$; $k_S$ denotes the saturation constant, $[g \cdot L^{-1}]$; and $Y_{S/X}$ denotes the yield coefficient, expressing the mass of cells formed per unit mass of consumed substrate, $[g \cdot g^{-1}]$.

The model is based on the following *a priori* assumptions [3]: (1) the bioreactor is completely mixed; (2) the main product is biomass; (3) the substrate glucose is consumed mainly oxidatively, and its consumption is described by Monod kinetics; (4) variations in the growth rate and substrate consumption do not change significantly the elemental composition of biomass, and thus, balanced growth conditions are assumed; (5) parameters (e.g., temperature, pH, $pO_2$) are controlled at their individual constant set points.

### 2.1.2. *S. cerevisiae* Fed-Batch Fermentation Process

A fed-batch fermentation process of *S. cerevisiae* has been carried out in the Institute of Technical Chemistry, University of Hannover, Hannover, Germany [3]. The available experimental data consist of on-line measurements of substrate (glucose) and off-line measurements of biomass and ethanol.

The on-line glucose measurements were performed by using two pumps (ACCU FM40, SciLog, Saint. Louis, MO, USA) for carrier and sample streams with flow rates of 1.7 mL·min$^{-1}$ and 1.0 mL·min$^{-1}$, respectively. Using two injection valves (Knauer, Berlin, Germany), 35 L glucose oxidase (Fluka, Seelze, Germany) solution (100,000 U·L$^{-1}$) as well as 18 L culture broth were injected in fast succession in the carrier stream, so that the culture broth would be placed in front of the enzyme solution in the carrier stream. A 50-cm-long tube with an inside diameter of 0.5 mm was employed as a manifold. The control of the FIA system and the measurement evaluations were performed by the automation system CAFCA (ANASYSCON, Hannover, Germany).

Roughly, about 10 mL of sample were taken every hour in order to determine the biomass and ethanol concentrations. They were immediately cooled down in ice water to prevent the metabolic activity of the cells in the sample. Ethanol was measured by the gas chromatograph Shimadzu GC-14B (Shimadzu, Duisburg, Germany).

Figure 2 presents the feeding profile for the *S. cerevisiae* fed-batch FP.

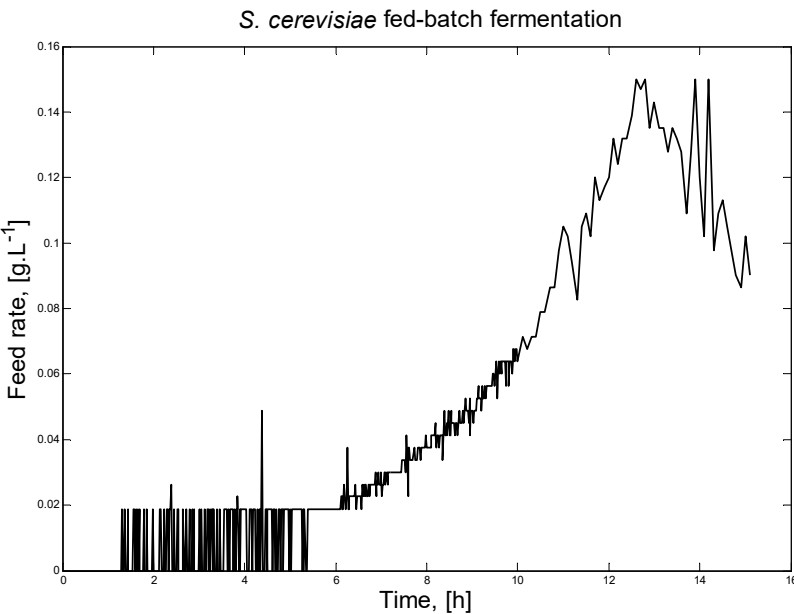

**Figure 2.** *S. cerevisiae* fed-batch FP feeding profile.

A full description of the process conditions and experimental data of the fermentation are given in [3].

The *S. cerevisiae* fed-batch FP is modelled by a system of four differential equations, which describes the nonlinear dynamics of biomass, substrate, and ethanol, considering the mixed oxidative functional state [3]:

$$\frac{dX}{dt} = \left(\mu_{2S}\frac{S}{S+k_S} + \mu_{2E}\frac{E}{E+k_E}\right)X - \frac{F}{V}X \tag{4}$$

$$\frac{dS}{dt} = -\frac{\mu_{2S}}{Y_{S/X}}\frac{S}{S+k_S}X + \frac{F}{V}(S_{in} - S) \tag{5}$$

$$\frac{dE}{dt} = -\frac{\mu_{2E}}{Y_{E/X}}\frac{E}{E+k_E}X - \frac{F}{V}E \tag{6}$$

$$\frac{dV}{dt} = F \tag{7}$$

where additionally to the aforementioned symbols, $E$ denotes ethanol concentration, $[g{\cdot}L^{-1}]$; $\mu_{2S}$, $\mu_{2E}$ denotes substrate and ethanol maximum growth rates, $[h^{-1}]$; $k_E$ denotes ethanol saturation constant, $[g{\cdot}L^{-1}]$; and $Y_{E/X}$ denotes yield coefficient, expressing the mass of cells formed per unit mass of ethanol consumed, $[g{\cdot}g^{-1}]$.

All functions in both models (Equations (1)–(3) for the *E. coli* FP and Equations (4)–(7) for the *S. cerevisiae* FP) are continuous and differentiable. The vectors of the parameters that are going to be identified are $p_1 = [\mu_{max}, k_S, Y_{S/X}]$ and $p_2 = [\mu_{2S}, \mu_{2E}, k_S, k_E, Y_{S/X}, Y_{E/X}]$, respectively for each FP. The non-zero division requirement is fulfilled for the model parameters.

The following assumptions are made for the *S. cerevisiae* FP model [32]: (1) the main by-products in an aerobic yeast growth process are water, carbon dioxide, and ethanol; (2) the bioreactor is completely mixed; (3) ethanol consumption is inhibited when sugar concentration in the broth is higher than a critical level; (4) the elemental composition of yeast in the process does not significantly change; (5) parameters, e.g., pH and temperature, are controlled to certain acceptable constant values during the process.

Taking into account the fermentation conditions, the *S. cerevisiae* FP falls in the mixed oxidative state [3,32]. The specific growth rate is expressed as a sum of two terms (Equation (4)), one describing the contribution of sugar and the other describing the contribution of ethanol to yeast growth. Both terms have the structure of a Monod model. A Monod model is also used for the specific ethanol (Equation (6)) and sugar consumption rates (Equation (5)).

Both FPs studied here are chosen so as to explore the behaviour of the metaheuristic algorithms, starting with a simple mathematical model (three model parameters for identification, Equations (1)–(3)), followed by a more complicated model (six model parameters, Equations (4)–(7)).

The deviation between the modelled and experimental FP data is defined as an optimisation criterion:

$$J = \|Z\|^2 \to min \tag{8}$$

where $\|\ \|$ denotes the $\ell^2$–vector norm, and $Z = Z_{mod} - Z_{exp}$, $Z_{mod} \stackrel{def}{=} [X_{mod}\ S_{mod}]$ (*E. coli* FP model), or $Z_{mod} \stackrel{def}{=} [X_{mod}\ S_{mod}\ E_{mod}]$ (*S. cerevisiae* FP model) are respectively the biomass, substrate, and ethanol model predictions, interpolated in the time scale of available experimental data (on-line or off-line). Respectively, $Z_{exp} \stackrel{def}{=} [X_{exp}\ S_{exp}]$ or $Z_{exp} \stackrel{def}{=} [X_{exp}\ S_{exp}\ E_{exp}]$ are known experimental data for the process variables.

*2.2. Metaheuristic Algorithms*

2.2.1. Genetic Algorithm

The individuals employed in GA, also known as chromosomes, are sets of coded parameters.

Solution representation. A problem's solution is emulated as one artificial chromosome. Each chromosome is a set of genes or a string over a definite alphabet. Most often, genes are encoded as 0 s or 1 s. The genes in a chromosome contain information for the corresponding problem's parameters.

In search of a global optimal solution, the simple genetic algorithm works with one population. Fundamental genetic operators as selection, crossover, and mutation are exploited in the process.

Selection. The selection operator is used for choosing those chromosomes that are better possible solutions. The comparison is based on the values of the objective function or the fitness $F_i$ of each chromosome $i$. The most commonly used selection method is the roulette wheel selection. The probability $P_i$ for selecting the individual $i$ is defined as:

$$P_i = \frac{F_i}{\sum\limits_{j=1}^{N_{pop}} F_j} \tag{9}$$

where $N_{pop}$ is the size of the population.

Reproduction. Crossover and mutation operators are employed during the reproduction phase. A new offspring is generated by the crossover operator. Mutation occurs next to avoid falling into a local optimum.

Let the pair of parents selected to breed be presented by a pair of $D$–dimensional vectors $\overline{X}$ and $\overline{Y}$, where $D$ is the dimension of the search space. For binary $\overline{X}$ and $\overline{Y}$, a simple crossover and binary mutation are:

$$x'_i = \begin{cases} x_i, & \text{if } i < r \\ y_i, & \text{otherwise} \end{cases}, \quad y'_i = \begin{cases} y_i, & \text{if } i < r \\ x_i, & \text{otherwise} \end{cases} \tag{10}$$

$$x_i = \begin{cases} 1 - x_i, & \text{if } U(0, 1) < p_m \\ x_i, & \text{otherwise} \end{cases} \tag{11}$$

where $p_m$ is the mutation probability (binary) and $r$ is a random number.

The influence of the GA parameters on the algorithm's performance is shown in the so-called reproductive scheme growth equation (fundamental theorem of genetic algorithms) [9]:

$$\xi(S, t+1) \geq \xi(S, t) \times \frac{l(S, t)f(S)}{\overline{F}(t)} \left[ 1 - p_c \times \frac{\delta(S)}{m-1} - o(S) \times p_m \right] \tag{12}$$

where $l(S, t)$ denotes the number of strings from generation $t$ that belong to the schema $S$; $f(S)$ denotes the average fitness of the schema $S$; $\overline{F}(t)$ denotes the average fitness of the population from generation $t$; $p_c$ denotes the probability of crossover; defining length $\delta(S)$ denotes the distance between the first and last specific positions; $o(S)$ denotes the order of the schema; and $m$ denotes the length of the code.

GA terminates upon reaching a certain stop criterion, e.g., completion of a predefined number of iterations.

The block chart of the GA, implemented and tuned for the identification of model parameters of the fed-batch fermentation processes studied here, is presented in Figure 3.

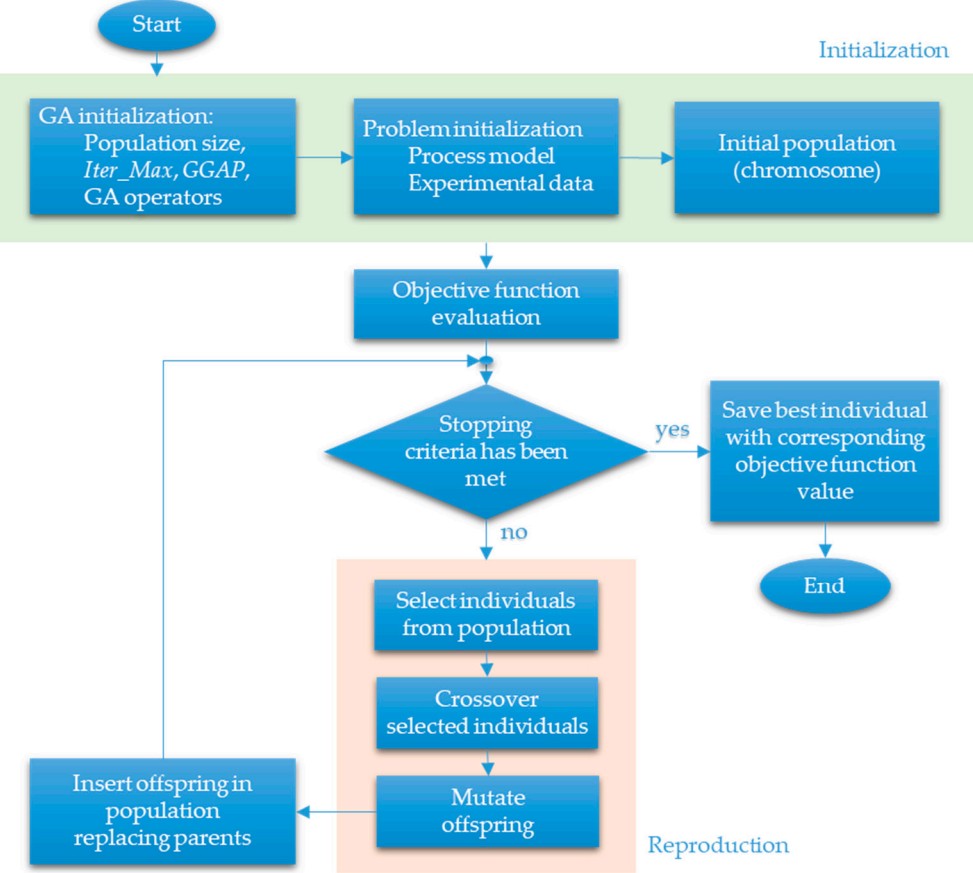

**Figure 3.** Block chart of genetic algorithm (GA).

### 2.2.2. Water Cycle Algorithm

Raindrops in case of rain or precipitation are considered the WCA initial population. The best individual (*raindrop*) is regarded as a sea. The rivers are represented by the worthy raindrops. The remaining raindrops are the streams that run into the rivers and the sea. The block chart of WCA is presented in Figure 4.

A *raindrop* or a single solution of the optimisation problem under consideration in terms of WCA is a $1 \times N_{\text{var}}$ dimensional array, which is defined as:

$$Raindrop = [X_1, X_2, \ldots, X_{N_{\text{var}}}] \tag{13}$$

where $N_{\text{var}}$ is the dimension of the optimisation problem.

The cost function (*Cost*) of the raindrop could be determined by:

$$Cost_i = \int \left(X_1^i, X_2^i, \ldots, X_{N_{\text{var}}}^i\right), i = 1, 2, 3, \ldots, N_{pop} \tag{14}$$

where the count of the raindrops is stored in $N_{pop}$.

Depending on the strength of the flow, the following equation controls whether the raindrops flow into the sea or into the rivers:

$$NS_n = round\left\{ \left| \frac{Cost_n}{\sum_{i=1}^{N_{sr}} Cost_i} \right| \times N_{Raindrops} \right\}, n = 1, 2, \ldots, N_{sr} \tag{15}$$

where $N_{sr}$ is the total number of rivers and a single sea, and $N_{Raindrops}$ is the rest of the population (raindrops flowing straight into the sea or into the rivers).

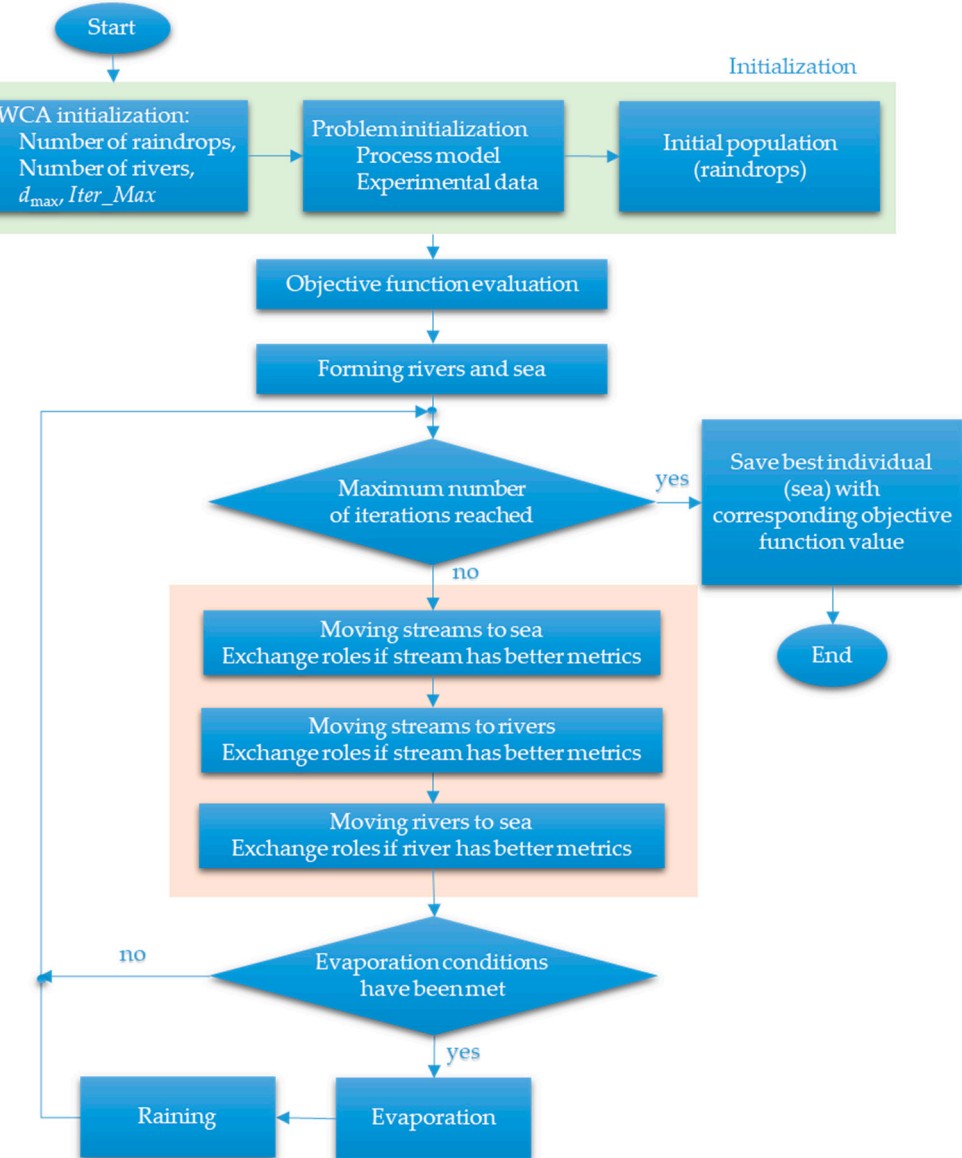

**Figure 4.** Block chart of the water cycle algorithm (WCA).

The raindrops create the streams. The streams link to generate the rivers or flow straight into the sea. The sea (the best optimal point) is the ultimate end point of all rivers and streams. The new position for the streams and rivers is evaluated by:

$$X_{Stream}^{i+} = X_{Stream}^i + rand \times C \times (X_{River}^i - X_{Stream}^i) \tag{16}$$

$$X_{River}^{i+} = X_{River}^i + rand \times C \times (X_{Sea}^i - X_{River}^i) \tag{17}$$

where $C$ represents a value between 1 and 2, and *rand* is a uniformly distributed random number in (0, 1).

The roles of a stream and the river it flows into can be exchanged if the solution represented by the stream is better than the one represented by the river. A similar swap of roles is also possible for the sea and the rivers.

The value of the parameter $d_{\max}$ determines if additional search around the sea (the optimum solution) should be prevented or encouraged:

$$d_{\max}^{i+1} \; = \; d_{\max}^{i} - \frac{d_{\max}^{i}}{Iter\_Max}.$$  (18)

Rain occurs after evaporation. New streams are composed by the new raindrops in different locations specified as:

$$X_{Stream}^{new} \; = \; LB \, + \, rand \, \times (UB \, - \, LB)$$  (19)

where *UB* and *LB* are respectively the upper and lower bounds of the search parameters defined for the given problem.

Equation (20) is used for those streams that flow straight into the sea to improve the algorithm's performance and the convergence for constrained problems:

$$X_{Stream}^{new} \; = \; X_{sea} + \; \sqrt{\mu} \times randn \, (1, \, N_{var})$$  (20)

where the coefficient $\mu$ indicates the range of the search area close to the sea, $\sqrt{\mu}$ is a standard deviation, and *randn* is a normally distributed random number. It is worth noting that it is more likely that the feasible area will be left at a higher value of $\mu$, while a lower value of $\mu$ pushes the algorithm to search in a tight area close to the sea. The coefficient $\mu$ can be regarded as the concept of variance, meaning that the individuals generated with variance $\mu$ are distributed around the best optimum point (the sea).

## 3. Results and Discussion

### 3.1. Algorithms Implementation for Model Parameter Identification

The following case studies are the subject of this investigation:

- *Case study 1*: Parameters identification of an *E. coli* fed-batch FP model, presented by Equations (1)–(3) with an optimisation criterion expressed by Equation (8); and
- *Case study 2*: Parameters identification of a *S. cerevisiae* fed-batch FP model, presented by Equations (4)–(7) with an optimisation criterion expressed by Equation (8).

GA and WCA are adapted for the parameter identification of the two case studies considered here. All computations are run using a PC/Intel Core i5-2320 CPU @ 3.00 GHz, 8 GB Memory (RAM), Windows 10 (64 bit) operating system.

The implementation of both algorithms is performed in a Matlab R2013a environment. GA is implemented based on the functions proposed in [33], while the WCA implementation follows the standard version of WCA proposed in [20].

It is known that the stochastic algorithms, such as GA and WCA discussed here, return different results for different runs of the same algorithm on the same input dataset. When comparing algorithms, it is recommended to use the average results to evaluate the algorithm performance. It is often advised to use at least 30 independent algorithm runs. Therefore, GA and WCA have been executed 30 times without any change in both algorithms' parameters and functions.

In order to achieve the best algorithm performance in terms of solution quality (higher model accuracy, i.e., a lower objective function value, for appropriate estimates of the model parameters), adjustments of GA and WCA parameters and functions are made. The parameters of both metaheuristic algorithms are tuned on the basis of a series of pre-tests and according to the domains of the problems considered here. The parameters chosen for both algorithms are listed in Table 1.

**Table 1.** GA and WCA parameters.

| Parameter | GA | WCA |
|---|---|---|
| Population number $\left(N_{pop}\right)$ | 200 | 200 |
| Number of rivers plus sea $(N_{sr})$ | - | 50 |
| Generation gap (*GGAP*) | 0.97 */0.8 ** | - |
| Maximal number of iterations (*Iter_Max*) | 400 | 400 |
| Crossover probability $(p_c)$ | 0.7 */0.95 ** | - |
| Mutation probability $(p_m)$ | 0.05 */0.1 ** | - |
| Evaporation condition constant $(d_{\max})$ | - | $1 \times 10^{-15}$ |

*\* Case study 1; \*\* Case study 2.*

### 3.2. Case Study 1: E. coli Fed-Batch Fermentation Process

The observed objective function values *J* and the estimates of the model parameters $\mu_{\max}$, $k_S$ and $Y_{S/X}$ are collected. The comparison of the results is made based on the obtained best *J* (lowest *J* value), worst *J* (highest *J* value), and mean *J* (average *J* value) values from the performed 30 runs. The average computational time is about 1.5 min/run for GA and approximately 5 min/run for WCA. Since the problem of a model parameter identification is an off-line task, the computational time is not the most important feature to be considered. The most significant one is the achieved model accuracy, which is the main challenge for the metaheuristic algorithms applied here.

The numerical outcomes are listed in Table 2, where the values of model parameters correspond to the algorithm run with the reported *J* value. In addition, the parameter estimations are enhanced by presenting the 99.9% confidence intervals (CI) and the standard deviation (SD).

**Table 2.** *Case study 1*: Comparison of the results obtained by GA and WCA. CI: Confidence intervals.

| Parameter | GA | | | | WCA | | | |
|---|---|---|---|---|---|---|---|---|
| | **Best** | **Worst** | **CI** | **SD** | **Best** | **Worst** | **CI** | **SD** |
| *J* | 4.392 | 4.535 | 4.468 ± 0.0218 | 0.0363 | 4.222 | 4.477 | 4.352 ± 0.0437 | 0.0727 |
| $\mu_{max}$, [h$^{-1}$] | 0.489 | 0.493 | 0.4898 ± 0.0018 | 0.0030 | 0.478 | 0.501 | 0.4875 ± 0.0054 | 0.0091 |
| $k_S$, [g·L$^{-1}$] | 0.012 | 0.013 | 0.0123 ± 0.0003 | 0.0005 | 0.010 | 0.015 | 0.0116 ± 0.0010 | 0.0017 |
| $Y_{S/X}$, [g·g$^{-1}$] | 2.019 | 2.019 | 2.0206 ± 0.0007 | 0.0012 | 2.018 | 2.019 | 2.0194 ± 0.0006 | 0.0011 |

The results, presented in Table 2, show that the highest quality solution, corresponding to the lowest objective function value (*J* = 4.222), is found by WCA. The observed mean value of *J* = 4.352 obtained by WCA is better than the best *J* value obtained by GA (*J* = 4.392), while the mean *J* value of GA (*J* = 4.468) is closer to the worst *J* value of WCA (*J* = 4.477). The numerical results definitely confirm the better performance of WCA compared to GA.

The distribution of both algorithms' estimates of the model parameters is discussed based on exploratory graphics as box plots, which is a standard technique for analysis of the obtained results. The obtained box plots are presented in Figure 5. It is shown that the variability of WCA samples is bigger than the GA variability. The only exceptions are the estimates obtained for the parameter $Y_{S/X}$ (Figure 5d). This is an interesting result, taking into account that the parameter $Y_{S/X}$ is the least sensitive one of the model parameters in *Case study 1* [34]. The sensitivity of $\mu_{max}/k_s$ ratio, referring to the kinetics of enzymatic substrate uptake and microbial growth, is worth considering when identifying parameters of nonlinear growth models of the Monod type [35].

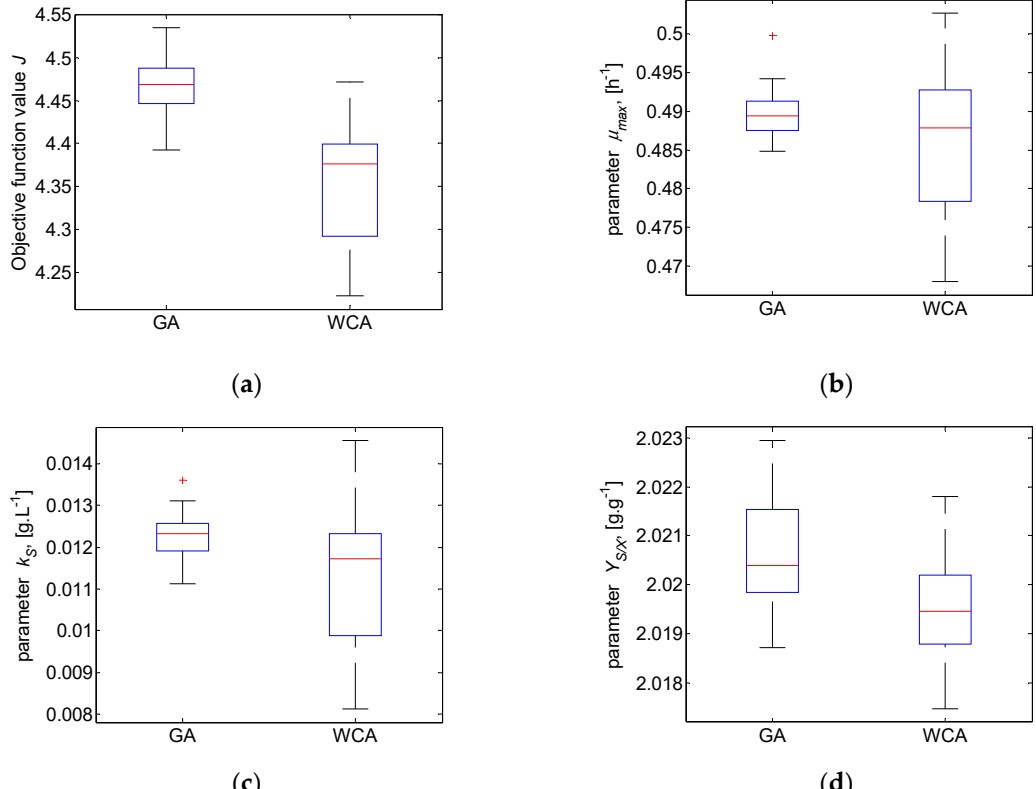

**Figure 5.** *Case study 1:* Box plot for: (**a**) objective function value *J* and model parameters (**b**) $\mu_{\max}$; (**c**) $k_S$; (**d**) $Y_{S/X}$.

The results presented in Figure 5a show that the worst *J* value obtained by WCA (highest *J* value of 4.477) is very close in value to the median (red line) of the GA results (average *J* value of 4.468), and the best *J* value obtained by GA (lowest *J* value of 4.392) is very close in value to the median of the WCA results (average *J* value of 4.352). Although the WCA results show bigger variability, they are more accurate than the GA results. The results for the two most sensitive model parameters $\mu_{\max}$ and $k_S$ (Figure 5b,c) are similar—medians are close in value, but the obtained WCA estimates are much more diverse. It is noteworthy that the values of the parameters for which the smallest error is achieved by WCA are far beyond the highest and the lowest GA results (upper and lower marks of the GA box plot diagram). It can be seen that WCA covers most of the problem parametric space.

Among GA estimates, one outlier is observed for each parameter $\mu_{\max}$ and $k_S$. Such a result could be explained by the observed good exploration of GA while searching for solutions in new regions, against the weaker exploitation. The better performance of WCA, i.e., the obtained lowest *J* value, might be related to the observed better WCA exploitation (the refinement of already existing solutions) compared to the GA exploitation. This finding would be of interest for future research.

A graphical representation of the *E. coli* fed-batch FP model outcomes (biomass and substrate) and the real experimental data is presented in Figure 6 to clearly show the existence of systematic deviations between the model predictions and the measurements. A part of the graphics showing the difference between GA and WCA results is zoomed in order to be presented more clearly.

The comparison of biomass time profiles (experimental and predicted data) obtained by GA and WCA is presented in Figure 6a. Model predictions of the biomass concentration for the estimated model parameters of the best performance both of GA and WCA are in accordance with the measured experimental data. Although both algorithms fit the experimental data very well, WCA predicts the data with a higher degree of accuracy. The same behaviour is demonstrated in Figure 6b, where the dynamics of the modelled and measured substrate are compared.

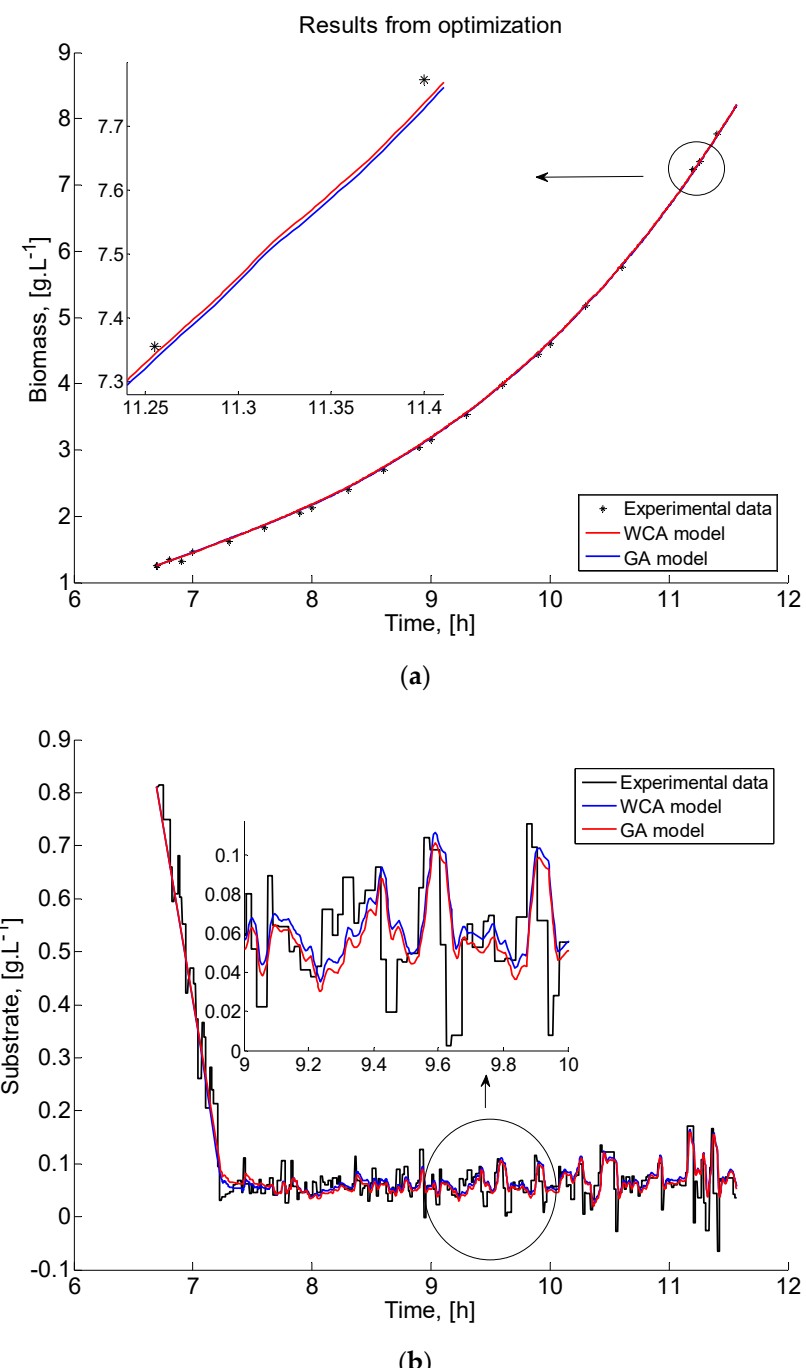

**Figure 6.** *Case study 1*: Comparison of the process variables time profiles obtained by GA and WCA: (**a**) biomass; (**b**) substrate.

The presented graphical results are a successful validation of the adequacy of the obtained models.

### 3.3. Case Study 2: S. cerevisiae Fed-Batch Fermentation Process

In analogy to *Case study 1*, GA and WCA have been run 30 times for the same set of algorithms' parameters and functions. Table 3 presents the best and the worst values of the objective function $J$ and the model parameters $\mu_{2S}$, $\mu_{2E}$, $k_S$, $k_E$, $Y_{S/X}$, and $Y_{E/X}$, as well as the 99.9% CI and SD for each of them. The results are based on the performed 30 runs, ensuring a fair comparison of the stochastic algorithms discussed here.

**Table 3.** *Case study* 2: Comparison of the results obtained by GA and WCA.

| Parameter | GA | | | | WCA | | | |
|---|---|---|---|---|---|---|---|---|
| | Best | Worst | CI | SD | Best | Worst | CI | SD |
| $J$ | 1.338 | 5.107 | 1.508 ± 0.4299 | 0.7155 | 1.326 | 1.371 | 1.345 ± 0.0081 | 0.0134 |
| $\mu_{2S}$, [h$^{-1}$] | 0.839 | 1.000 | 0.9748 ± 0.0223 | 0.0371 | 0.594 | 1.000 | 0.8284 ± 0.1042 | 0.1735 |
| $\mu_{2E}$, [h$^{-1}$] | 0.093 | 0.224 | 0.1129 ± 0.0143 | 0.0239 | 0.050 | 0.124 | 0.0930 ± 0.0146 | 0.0243 |
| $k_S$, [g·L$^{-1}$] | 0.074 | 0.143 | 0.1180 ± 0.0066 | 0.0110 | 0.050 | 0.128 | 0.0928 ± 0.0193 | 0.0321 |
| $k_E$, [g·L$^{-1}$] | 0.472 | 0.995 | 0.8762 ± 0.0691 | 0.1150 | 0.128 | 1.000 | 0.6918 ± 0.1896 | 0.3155 |
| $Y_{S/X}$, [g·g$^{-1}$] | 2.093 | 2.276 | 2.2222 ± 0.0158 | 0.0262 | 2.224 | 2.231 | 2.2275 ± 0.0011 | 0.0019 |
| $Y_{E/X}$, [g·g$^{-1}$] | 1.202 | 4.459 | 1.3956 ± 0.3498 | 0.5821 | 1.169 | 1.300 | 1.2389 ± 0.0210 | 0.0350 |

The numerical estimates presented in Table 3 show that the objective function mean value of 1.345 obtained by WCA is considerably lower compared to the GA *J* mean value of 1.508. The results demonstrate the better performance of WCA. In *Case study 2* (the more complex problem), WCA again finds the solution with the highest quality, i.e., the solution with the lowest value of *J* = 1.326. If one compares the best values of both algorithms, the difference between them (*J* = 1.326 for WCA and *J* = 1.338 for GA) is not as significant as observed in *Case study 1*. It should be noted that considering the *J* mean value, the one obtained by WCA is much lower than the *J* mean value of GA. These results distinguish WCA as more appropriate in the case of a complex model with 6 parameters as well.

Following the analysis performed in *Case study 1*, the parameter estimates obtained by the two algorithms are discussed again based on the exploratory graphics as box plots. Figure 7 presents the graphics successively for the objective function *J* and the model parameters. In this *Case study 2*, the variability of WCA samples is again bigger than the GA variability.

The results from Figure 7a and Table 3 show that WCA surpasses the performance of GA—even the worst *J* value obtained by WCA is below the value of the median of the GA results. Comparing the best values of the objective function, the results obtained by GA and WCA are very close—1.33 versus 1.34. However, if one compares the mean values, which is recommended in cases of metaheuristic algorithms, WCA outperforms GA with 1.35 versus 1.51, which is almost a 12% increase in the model accuracy. In general, it is again demonstrated that WCA covers a broader parametric space than GA. The results obtained for the model parameters $\mu_{2S}$, $k_S$, and $k_E$ (Figure 7b,d,e) look similar—despite the very close values obtained for the objective function *J*, different values are recorded for the parameter estimates, with bigger deviations in case of WCA implementation. The results obtained for the model parameters $\mu_{2E}$, $Y_{S/X}$, and $Y_{E/X}$ (Figure 7c,f,g) also look similar—there are closer values of the estimated parameters and a closer distribution of the estimates.

From another point of view, only GA shows outliers, but never WCA, as in *Case study 1*. In this more complex optimisation task, the weaker GA exploitation is even more evident—there are solutions found in more distant regions of the parameter space that remain unexplored. At the same time, WCA preserves its promising efficiency, finding the solution with the highest quality, i.e., the lowest *J* value. In order to understand more thoroughly the nature of the observed outliers, further research is necessary.

Figure 8 demonstrates the dynamic behaviour of the three process model variables (biomass, substrate, and ethanol) and the corresponding experimental data of the *S. cerevisiae* fed-batch FP. Analogously to *Case study 1*, a part of the graphics is zoomed to show the difference between GA and WCA results more clearly, even though in this case, it is obvious.

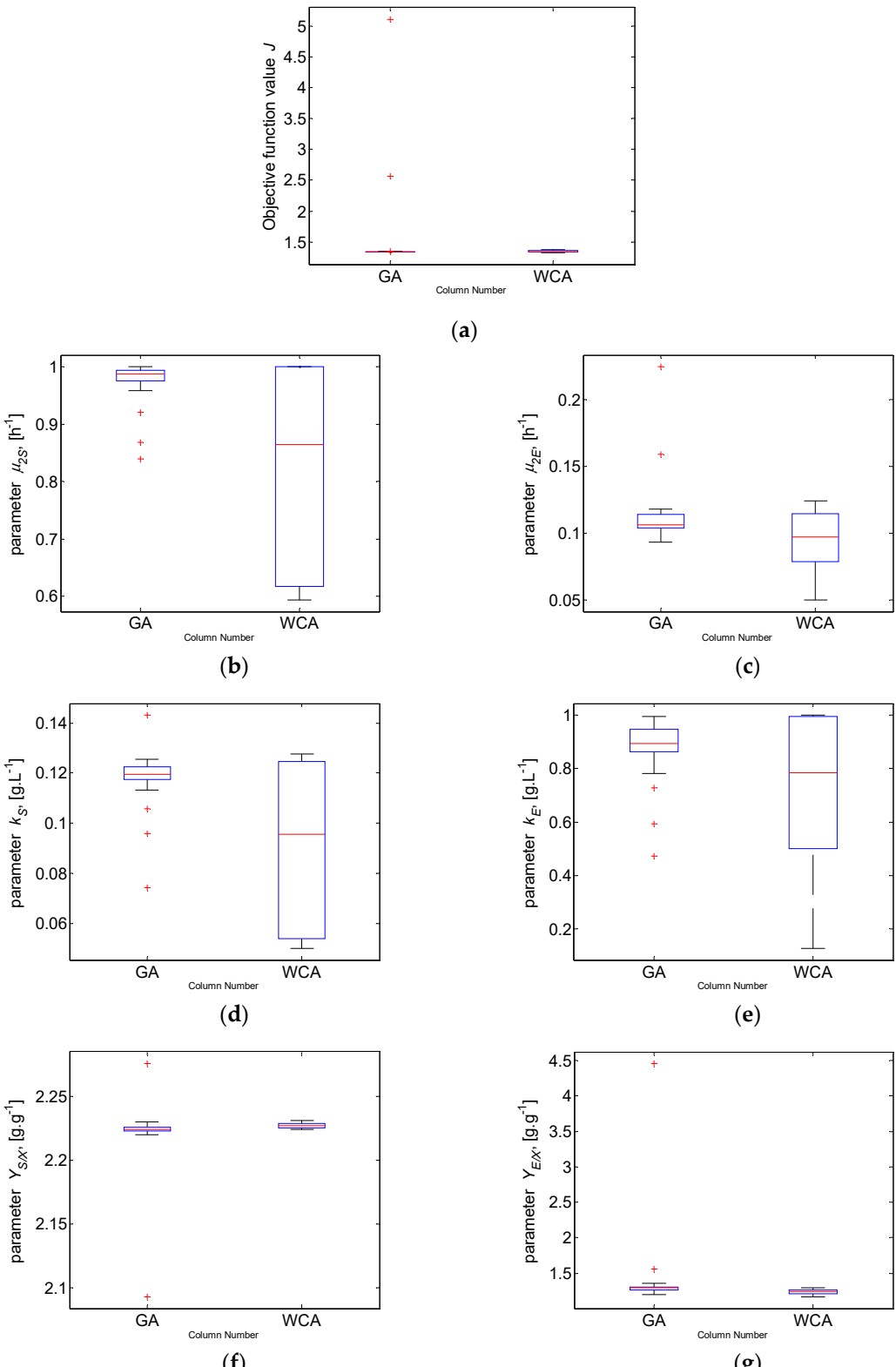

**Figure 7.** *Case study 2*: Box plot for: (**a**) objective function value *J* and model parameters (**b**) $\mu_{2S}$; (**c**) $\mu_{2E}$; (**d**) $k_S$; (**e**) $k_E$; (**f**) $Y_{S/X}$; (**g**) $Y_{E/X}$.

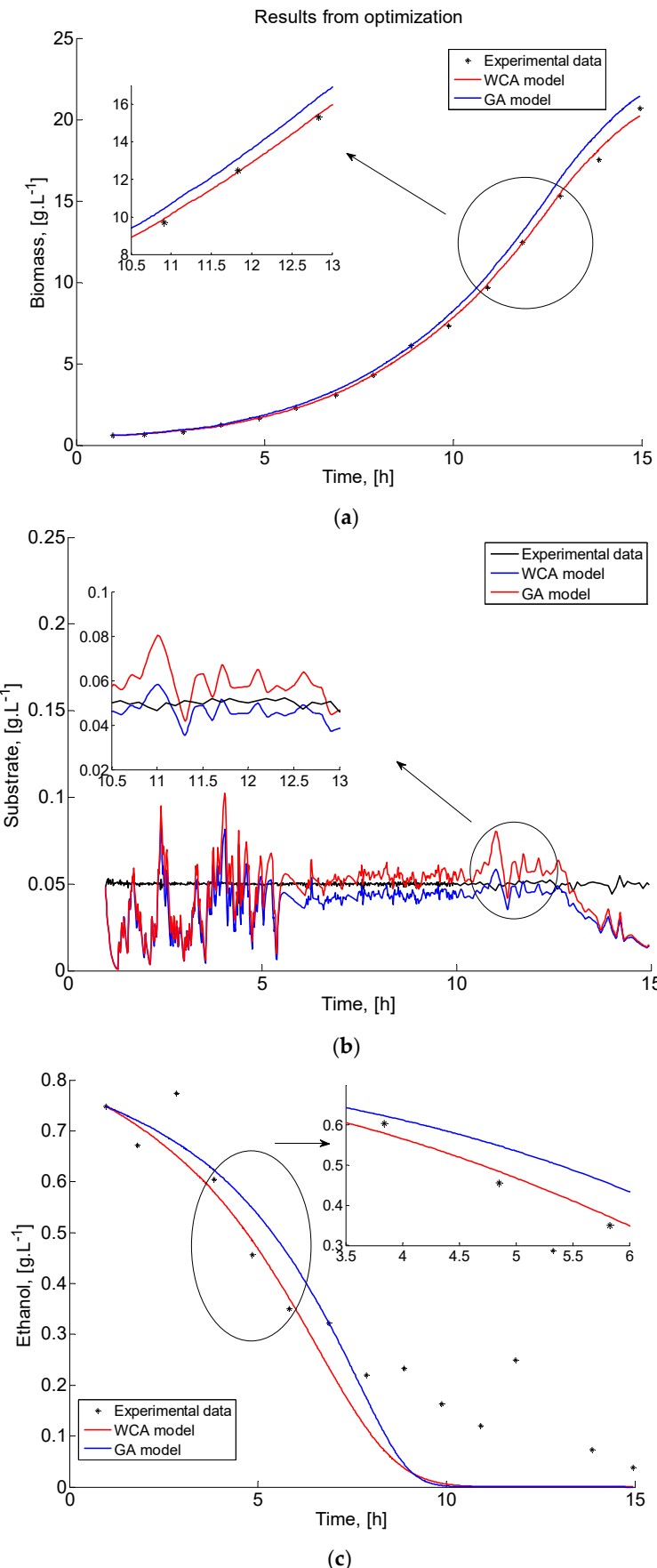

**Figure 8.** *Case study 2*: Comparison of the process variables time profiles obtained by GA and WCA:
(**a**) biomass; (**b**) substrate; (**c**) ethanol.

As seen in Figure 8, both GA and WCA models predict well the experimental data for the three process variables. However, WCA visually fit the data with a higher degree of accuracy considering biomass, substrate, and ethanol dynamics, as confirmed by the numerical results presented in Table 3.

## 4. Conclusions

Throughout this work, the feasibility of the metaheuristic water cycle algorithm applied to a parameter identification problem has been highlighted. WCA has been adapted and implemented here for the first time for an estimation of FP models parameters. To verify the WCA effectiveness, a series of identification procedures of two different FP models has been carried out. The bacteria *E. coli* and the yeast *S. cerevisiae* have been chosen as case studies.

To confirm the superiority of WCA, a comparison has been made with the results obtained by GA. It has been shown that WCA outperforms GA—the observed WCA model accuracy is better than the accuracy achieved by GA for both models, the simple one (*E. coli*) and the more complex one (*S. cerevisiae*). In addition, the numerical results are studied using box plots. Based on the algorithms performance analysis, better exploitation of WCA has been observed in comparison to the GA exploitation. The adequacy of the obtained models is further successfully validated by a graphical representation of simulated model predictions compared to the available experimental data for both considered case studies. Thus, the efficiency of both metaheuristics is confirmed. It is shown that the better WCA performance is preserved even for the more complex model: The yeast fermentation process.

In summary, it has been demonstrated that WCA results are more accurate compared to GA results. Taking into account that WCA is simple in terms of coding, implementation, and its confirmed superiority, it could be concluded that WCA is an efficient and powerful algorithm for the parameter identification of complex nonlinear FP models.

The development of models that describe so adequately the aspects significantly affecting process performance is a crucial step for the further elaboration of high-quality control strategies in order to optimise fermentation production processes. Process models can be useful for more than controller design tasks. They can be used to synthesise software sensors (estimation of unmeasurable variables), as well as to simulate the process behaviour when investigating the system dynamics or testing a developed control law.

**Author Contributions:** Conceptualisation and methodology, O.R. and T.P.; software implementation, O.R., D.Z. and M.A.; validation, O.R. and T.P.; investigation, O.R., M.A., D.Z. and T.P.; writing—original draft preparation, O.R., M.A., and T.P.; writing—review and editing, O.R., M.A., D.Z. and T.P.; visualisation, O.R., D.Z. and T.P.; supervision, O.R. and T.P. All authors have read and agreed to the published version of the manuscript.

**Funding:** This research was funded by the Bulgarian National Science Fund, Grant DN02/10 "New Instruments for Knowledge Discovery from Data, and their Modelling".

**Conflicts of Interest:** The authors declare no conflict of interest.

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
