# Peer review of "Water Cycle Algorithm for Modelling of Fermentation Processes"

_processes, doi:10.3390/pr8080920_

Round 1

Reviewer 1 Report

The submitted manuscript may interest researchers involved in modelling of biologically catalyzed processes. It is generally well written and easily comprehensible. There are, however, some issues may be improved before publication:

  1. The first paragraph of 2.1. that explains the choice of the modelled processes would better fit the Introduction.
  2. The first paragraphs of Results and Discussion (lines 193-202) seem to be more fit to the Materials and Methods section.
  3. Labels of axes of the box plots should include units. This regards tables as well. 
  4. The manuscript should be checked for the correctness of language, especially the Introduction section. 

Reviewer 2 Report

- Why do the authors only compare with GA? Is there a logistic reason for that?

- Materials and methods: The authors MUST explain, at least briefly, which variables are measured and also how they are measured. I know that further details are given in ref [3]. That is fine. But here, at least, mention what is used to measure e.g. glucose. Also in what units.

- The authors mentioned that there are variables measured on-line and off-line. How did the authors synchronize the time gaps that will exist between on and off-line measurements?.

- Eq 8: The minimization problem is expressed in a very strange manner. Is it Frobenous norm?

- Line 193: Version of Matlab?

- Apart from the fact that these algorithms have an important number of parameters to be optimized (and thus a high number of samples needed) I completely miss the validation step. How am I sure that the parameters found by the algorithm are the proper ones? This must be done by a comprehensive validation. Unless this validation is performed, the manuscript cannot be accepted.

Reviewer 3 Report

The present manuscript deals with the application of a new parameter estimation algorithm (i.e., water cycle algorithm) in well-known fermentation processes. The objective of the authors is to compare the accuracy of the algorithm with the accuracy of the already established genetic algorithm. In general, this issue is really interesting and worthy of studying, since the applications of fermentation models are numerous. By following a well-defined methodology, the authors conclude that the WCA can deliver predictions of better quality. In principle, this is true and supported by the results, though is in not generic enough. The comparison of the simulations with the experimental data is missing from the manuscript. Moreover, there is no discussion about the large variety of available mathematical model for fermentation processes (i.e., mechanistic, macroscopic, metabolic, kinetic, hybrid, etc.) and the potential integration of the WCA with them. In what follows, the weaknesses and shortcomings of the manuscript, which prevent its publication to the Processes Journal in its present form, are listed.

  • In Section 2.1.1, the model equations should be discussed in more detail. Which systems do they describe, what is the physical meaning of the parameters (e.g., the Y yields), do they consider inhibition of substrate and/or product mechanisms, are the cells able to grow both on the substrate and the product, etc.
  • The evaluation of the new WCA methodology is made by its comparison with the GA methodology to conclude that the former is able to estimate the parameters more accurately. The evaluation of WCA should be extended in order to prove that the model predictions are in well accordance with the experimental data.
  • In this kind of studies, it is crucial to present the comparison of the model predictions with the actual experimental data. As the authors have available data from existing fermentation processes they should produce comparison figures for the two algorithms. This approach should be based on the dynamic (time-dependent) comparison of the predictions of the two algorithms for biomass concentration, substrate concentration, ethanol concentration and volume.
  • The models for the two microorganisms are adapted under fed-batch conditions, though no details are provided for the feeding policies (except for the cited publications).
  • Some paragraphs of the Materials and Methods section belong better to the Introduction and should be transferred there. For example, the paragraph in lines 73-77, the paragraph in lines 110-112 and the paragraphs in lines 142-155.
  • Since the authors highlight it, the computational times for the two parameter estimation approaches should be presented and compared.
  • The sequence of the 30 model runs needs to be discussed in more detail. What changes were implemented from one run to the other? Why did the authors compare the average of all runs and not only the best one? Following the variability of the two algorithms, it is rather logic to continue only with the best estimation run for each case.
  • In Tables 2 and 3, the significance of the parameter estimation should be enhanced by including the confidence interval for each parameter.
  • In line 266, the statement “… the worst … results.” is not very clear from the relevant figure. It should be justified better.
  • In the case of GA methodology, there are a lot of outliers in the Box-plots. I understand that more research is needed to explain them, though they seem significant for the scope of the present manuscript.
  • The authors use the very generic term “solution quality”. This term should be explained better.
  • How did the authors conclude that the parameters μ_max and k_s are the most sensitive ones?
  • The English language quality of the manuscript is in general acceptable. Though, there are some grammatical and syntax errors scattered all over the text that need to be corrected. I suggest the text to be proof read once more by a native English speaker. Only some examples of the revisions that need to be made: in line 27, the sentence should be “… studying complex systems such as biological …”; in line 29, the sentence makes no sense and should be rephrased; in lines 37-38, the sentence should be further explained; in line 53 the word “a” is missing before “promising”; in line 95, the word “of” should be replaced with “on”; in lines 210 and 244 the verb should be in singular.
  • The Conclusions section should be improved. Presently, it shortly repeats the Introduction and Materials and Methods sections and, more importantly, highlights the key findings only in five lines. It should be also strengthened with the potential future applications of the developed model.

Round 2

Reviewer 1 Report

The issues pointed out by the Reviewers were addressed. The manuscript has been improved significantly and is now fit for publication.

Reviewer 2 Report

The authors have replied properly to my queries

Reviewer 3 Report

The revised manuscript is a significant improvement over its original form. The major weaknesses of the present study are no longer present. It was extremely necessary to include additional figures that present the comparison of the model predictions with the available experimental data. Moreover, the discussion section was significantly strengthened. I have no further comments except: (1) the embedded top titles in some figures should be removed; they repeat the relevant labels, (2) the predictability of the model in some cases in not optimal, e.g., Figure 8(c) at late fermentation times; the optimization could be part of further studies. In this regard, the revised manuscript can be accepted for publication to Processes journal.